# Mobile Telemedicine Screening for Diabetic Retinopathy Using Nonmydriatic Fundus Photographs in Burgundy: 11 Years of Results

**DOI:** 10.3390/jcm11051318

**Published:** 2022-02-27

**Authors:** Anthony Charlot, Florian Baudin, Mélanie Tessier, Sarah Lebrize, Victoire Hurand, Déborah Megroian, Louis Arnould, Inès Ben-Ghezala, Alain Marie Bron, Pierre-Henry Gabrielle, Catherine Creuzot-Garcher

**Affiliations:** 1Ophthalmology Department, University Hospital, 14 Rue Paul Gaffarel, 21079 Dijon, France; anthony.charlot@chu-dijon.fr (A.C.); meltessier@hotmail.fr (M.T.); sarah.lebrize@chu-dijon.fr (S.L.); victoire.hurand@chu-dijon.fr (V.H.); deborah.megroian@chu-dijon.fr (D.M.); louis.arnould@chu-dijon.fr (L.A.); ines.ben-ghezala@chu-dijon.fr (I.B.-G.); alain.bron@chu-dijon.fr (A.M.B.); pierrehenry.gabrielle@chu-dijon.fr (P.-H.G.); catherine.creuzot-garcher@chu-dijon.fr (C.C.-G.); 2INSERM—Institut National de la Santé et de la Recherche Médicale, CIC 1432, Clinical Investigation Center, Clinical Epidemiology/Clinical Trials Unit, Burgundy University, 21000 Dijon, France; 3Physiopathologie et Epidémiologie Cérébro-Cardiovasculaires (PEC2, EA 7460), Burgundy University, 21000 Dijon, France; 4Eye and Nutrition Research Group, CSGA, UMR 1324 INRA, 6265 CNRS, Burgundy University, 21000 Dijon, France

**Keywords:** diabetic retinopathy, telemedicine, screening

## Abstract

We analyzed the results of mobile screening for diabetic retinopathy (DR) using retinal photographs, comparing these results between rural and periurban areas, and before and after the first national COVID-19 pandemic lockdown. The Burgundy Union Régionale des Professionnels de Santé (URPS) has organized an annual DR screening since 2004. The examination, performed by an orthoptist, consisted of taking the patient′s history, intraocular pressure measurement, and taking retinal photographs. After remote transmission, the examinations were interpreted by participating ophthalmologists at the Dijon University Hospital. In September 2016, the screening was open to periurban townships. In 11 years, 10,220 patients were screened: 1420 patients (13.9%) had DR of any type, with an average age of 68.5 (±11.3) years, and 59.2% were men. These patients had a statistically significantly higher glycated hemoglobin level (7.4% vs. 7.0%) and a longer duration of diabetes (13.8 vs. 9.3 years) than patients without DR. When comparing rural and periurban areas and periods before and after the beginning of the COVID-19 pandemic, we did not find any significant difference in the screening results. The results of this study are in line with the average findings of similar studies comparing screening strategies for DR. The early detection of DR can benefit from mobile telemedicine screening, identifying a considerable number of patients at an elevated risk, especially in rural areas where access to ophthalmological care is limited.

## 1. Introduction

Diabetes mellitus is a multifactorial disease with genetic and environmental causes whose prevalence continues to increase. Between 2000 and 2013, the rate of patients treated for diabetes in France increased by 5.4% per year [1]. In France, according to the International Diabetes Federation (IDF), the number of people between 20 and 79 years of age suffering from diabetes was estimated to be 3.48 million in 2019 (4.8% of the population). This number is expected to increase to 3.76 million in 2030 (5.5%) [2]. Worldwide, the number of people with diabetes is expected to grow from 463 million in 2019 to 642 million in 2040 [2]. This pathology causes multiple systemic complications, often of vascular origin, mainly linked to the duration of diabetes and glycemic control [3].

The most common ophthalmologic complication is diabetic retinopathy (DR): a microvascular complication that can lead to retinal ischemia, neovascular glaucoma, and thus functional or even anatomical loss of the eyeball [4,5]. Diabetes is the world′s leading cause of blindness in the working population [6]. The prevalence of DR of any type is estimated at around 30% of diabetic patients, increasing as the duration of diabetes, glycated hemoglobin level, blood pressure, and type 1 diabetes increases [7,8]. The screening of diabetic patients is essential for diagnosing DR at an early stage, setting up appropriate treatment, and preventing irreversible vision loss [9]. Annual screening with dilated fundus examination is recommended, but it remains insufficient in France. According to the Haute Autorité de Santé (HAS, French National Authority for Health) in its 2010 report, only 50% of diabetic patients received an annual ophthalmological consultation and 72% within 2 years. As a result, screening with nonmydriatic retinal photographs and remote interpretation has been considered [10]. Various standardized recruitment protocols have been implemented to increase annual screenings, whether via a general practitioner or an endocrinologist, in hospitals [11,12,13] or drugstores [14], or even set up by local authorities [15]. Mobile screening is justified in Burgundy because of its vast territory and a density of ophthalmologists that is lower than the national average, at 6.65 ophthalmologists per 100,000 inhabitants, versus 8.8 per 100,000 in France (French Ministry of Health).

The main objective of this study is to describe the prevalence and characteristics of DR in a mobile screening campaign in Burgundy, France. The secondary objectives are to describe the prevalence of ophthalmologic pathologies not related to diabetes, to compare results between rural and periurban populations, and describe the results before and after the COVID-19 pandemic.

## 2. Materials and Methods

This was a retrospective, cross-sectional, and descriptive study based on screening data analysis performed between September 2009 and November 2020 in Burgundy, France. The study complied with the Declaration of Helsinki, and the locally appointed ethics committee gave approval for the research protocol.

### 2.1. Recruitment and Procedure

This itinerant screening for DR has been organized every year since 2004 by the Burgundy Union Régionale des Professionnels de Santé—Médecins Libéraux (URPS-ML, Regional Union of Health Professionals–Private Physicians). Each screening campaign begins in September and runs through June of the following year. A break is usually taken in December and January. In 2020, due to the COVID-19 pandemic and the first national lockdown in France, screening was suspended between 20 February and 6 October. In every visited township, departmental healthcare organizations identify diabetic patients who did not have any ophthalmologic acts coded in the previous 2 years. Then, a letter is sent to inform and invite patients to this free screening. Before the COVID-19 pandemic, patients were invited to participate in the screening without an appointment. Since 2020, patients have been required to make an appointment beforehand to optimize the number of people. Sometimes, practitioners asked the patient to bring the mobile screening device to the medical offices, but this took place for a minority of the screenings. The Burgundy region was visited in 2 years, resulting in a lapse of 2 years before the same location was revisited. Since September 2016, screening campaigns have been open to periurban areas, whereas only the most rural townships were previously screened. We compared periurban and rural areas between September 2016 and June 2019, according to a classification of the Agence Régionale de Santé de Bourgogne Franche Comté (Regional Health Agency of Burgundy Franche Comté), initially in 5 categories on 52 social and health criteria (including socioeconomic level, healthcare supply, general and premature mortality rates, and the age of the population). We classed townships into two main categories: rural and periurban townships. Because of the COVID-19 pandemic, we decided not to include the period beginning in September 2019 in this analysis to avoid recruitment bias.

A trained orthoptist performed the screening with mobile equipment. It consisted in collecting information about the administrative characteristics, the patient′s medical follow-up (referring ophthalmologist, last ophthalmological check-up, referring physician, and endocrinological follow-up), and personal history (type and duration of diabetes, antidiabetic and antihypertensive treatment, last glycated hemoglobin, and ophthalmological history). Then, an ophthalmological examination was performed. Intraocular pressure measurements were taken using the Canon TX-20P air tonometer pachymeter (Canon, Tokyo, Japan), and fundus retinal photographs were taken using a Canon NM CR-2 Plus Fundus Camera (Canon, Tokyo, Japan). For each patient, two photographs were taken: the first one was centered on the posterior pole and the second on the optic nerve. In case of poor-quality photographs, mydriatic drops were instilled in both eyes, and the photographs were retaken. Then, the images were transmitted via a web platform to the ophthalmology department of the Dijon University Hospital for remote interpretation. The interpretation was based on the ALFEDIAM classification (Association de langue française pour l’Étude du diabète et des maladies métaboliques (French Language Association for the Study of Diabetes and Metabolic Diseases), derived from the ETDRS classification (Early Treatment Diabetic Retinopathy Study) [16,17]. Ophthalmologists classified the fundus examinations in a standardized form: no DR, mild non-proliferative DR (NPDR), moderate NPDR, severe NPDR, or proliferative DR (PDR) [17]. Other suspected ophthalmologic diseases (e.g., macular edema, ocular hypertension (OHT), optic nerve head anomaly, choroidal nevus, age-related macular degeneration (AMD), central retinal vein occlusion (CRVO), and cataracts) were noted. OHT was defined as an intraocular pressure measured over 21 mmHg. The results were transmitted to the patients′ general practitioner and the declared endocrinologist. The patient received only advice on when to consult an ophthalmologist if needed. The ophthalmologist chosen by the patient received the report before the consultation.

### 2.2. Statistical Analysis

The worst eye for DR status and other pathologies was considered for the analysis. First, data were described with descriptive parameters (percentages, mean, and standard deviation). Comparative analyses were performed using a modified Student *t*-test for the quantitative variables. For the qualitative variables, the chi-square test was used as appropriate. Otherwise, the Fisher test was used. The level of significance was set at 0.05. Multivariate analyses for rural versus periurban comparisons were performed using logistic regressions.

## 3. Results

### 3.1. Description of the Study Population

Between September 2009 and November 2020, 10,220 patients received at least one screening examination, and 12,723 examinations were performed. At the time of their first screening examination, patients were on average 68.5 (±11.2) years old and reported their diabetes was ongoing for a mean 10.0 (±9.4) years. The mean glycated hemoglobin level was 7.1% (±1.2), based on the self-report of 9145 patients (89.48%). Other characteristics of these patients at their first screening examination are summarized in Table 1. Only 43.4% (4438 people) of the patients had not had an ophthalmologic check-up over the previous 2 years, a percentage rising to 95.1% of patients who had not had a check-up over the previous year (9723 patients). The mean time since the last consultation was 3.7 (±3.6) years based on the declaration of 9198 patients (90.0%) who were able to date their last consultation.

The number of new patients between 2010 and 2020 is shown in Figure 1, with the mean value of 891.7 (±295.1) of new patients per year. Data from 2009 with only 411 new patients are not shown in the graph because they cover screening only from September to December of that year. The year with the most patients screened was 2019, with 1540 new patients. Due to the COVID-19 pandemic, only 489 new patients were recruited in 2020, which is the lowest number in this study for a full year.

Between September 2009 and November 2020, 12,723 examinations were performed. Due to recurrent visits in townships, 8321 patients (81.4%) had only one examination, while 1899 (18.6%) had two or more examinations during the period studied. For 11 years of screening, an average of 1060 (±431) exams per year were performed.

### 3.2. Screening Results

The results of analysis regarding only the first exam of each patient are shown in Table 2 (see Figure 2A,B for retinal photographs example). Over the 11 years of the study, the rate of DR in at least one eye in all stages was 13.9% (1420 patients out of 10,220). This rate rises to 14.8% after excluding 590 patients whose photographs were unreadable (5.8% of the patients excluded, 1420 patients with DR out of 9630 patients). At the same time, we found other ophthalmologic conditions in at least one eye other than DR: media opacity (692 patients, 6.8%); late AMD (347 patients, 3.4%); choroidal nevus (287 patients, 2.8%; see Figure 2C); CRVO (20 patients, 0.2%); optic nerve head anomaly (418 patients, 4.1%); and OHT (799 patients, 7.8%).

### 3.3. Comparison of Patients with and without DR of Any Type

We compared patients with DR of any type with those without, and we obtained two groups of 1420 and 8210 patients, respectively, after excluding 590 (5.8%) patients with unreadable photographs. The results are shown in Table 3. Patients with DR of any type had significantly more type 1 diabetes (*p* < 0.001), higher glycated hemoglobin (*p* < 0.001), longer duration of diabetes (*p* < 0.001), more insulin treatment (*p* < 0.001), and fewer oral antidiabetics (*p* < 0.001). They also significantly had a shorter time since the last visit (*p* = 0.04) and more often reported an attending ophthalmologist (*p* < 0.01).

### 3.4. Differences between Rural and Periurban Areas

Regarding the comparison between rural and periurban areas, between September 2016 and June 2019, 3783 patients underwent at least one examination: 2753 patients living in rural townships and 1230 patients in urban or periurban townships according to the patients’ home addresses. In the univariate analysis, patients living in rural areas were significantly younger (*p* < 0.001) than patients living in urban areas. In addition, there was a significantly smaller proportion of mild non-proliferative DR in patients living in rural areas (*p* = 0.04). There was no significant difference regarding other stages of retinopathy or nondiabetic ophthalmologic pathologies (Table 4).

Age, sex, glycated hemoglobin levels, insulin treatment, having an attending ophthalmologist, and having mild non-proliferative DR or no DR were used as adjustment factors to perform multivariate analysis, as seen in Table 5. After multivariate analysis, only age was statistically different between the two groups, with patients in rural areas 1.4 years younger than in periurban areas (odds ratio: (0.67–2.18)).

### 3.5. Differences before and after the First Lockdown in France Related to the COVID-19 Pandemic

The screening was stopped at the beginning of the lockdown (on 17 March 2020) and started again in October 2020. We decided to compare two different populations: people who underwent screening just before the first lockdown (between 1 December 2019, and 29 February 2020; 367 patients) and those who had screening after screening resumed (between 1 October 2020, and 31 November 2020; 441 patients). Due to the short duration of this period, none of the patients had multiple examinations. The results of these analyses are shown in Table 6. Patients exhibited the same characteristics apart from age: people were significantly older before lockdown (70.3 ± 11.7 years old before vs. 69.1 ± 10.1 years old after, *p* = 0.04). There was no significant difference concerning the stage of the DR between the groups (*p* = 0.56).

## 4. Discussion

In this study, the prevalence of DR at all stages combined was 13.9% at the first examination of each patient. Moreover, the DR rate was lower than the disease’s known prevalence in the literature, around 30% (34.6% in the meta-analysis of Yau et al. in 2012) [7]. However, this rate is related to a diagnostic approach using a fundus exam performed by an ophthalmologist or ETDRS standard seven-field imaging [17]. The lower rate may also be explained by the regular in-office ophthalmologic follow-up of patients with known DR or diabetes with general complications, and therefore at greater risk of DR. It is also conceivable that patients who take the initiative to attend such screening are those who have more rigorous medical monitoring of their diabetes and better-controlled diabetes, and therefore fewer risk factors for DR, producing a “healthy worker effect”. We also can imagine that patients with severe diabetic ophthalmologic damage leading to symptoms, such as decreased visual acuity will more readily visita an ophthalmologic consultation or emergency department and will not participate in this type of screening campaign. Finally, our screening technique with two-field imaging is less precise than a fundus examination or standard seven-field imaging and may misdiagnose some cases of mild to moderate NPDR. Moreover, recent studies using ultra-wide field (UWF) imaging did not show significant differences between ETDRS standard seven-field imaging and UWF imaging. However, in screening, it is easier and more precise to use UWF imaging with only one nonmydriatic photograph than multiple photograph imaging [18].

Comparing populations with or without DR, patients with DR were significantly more likely to have type 1 diabetes, had a longer course of diabetes, had higher glycated hemoglobin, and mainly were treated with insulin. These characteristics are well known as risk factors for developing DR [7,19]. Patients with DR were less likely to be treated with oral antidiabetics: this may be due to the greater proportion of type 1 diabetic people in DR who are not treated with this type of medication. We found no differences in other discussed risk factors because it was not statistically significant (gender) or because data were not collected (smoking, blood pressure, and cholesterol levels) [19]. Surprisingly, patients with DR reported an attending ophthalmologist more often and had a shorter time since the last visit than patients without DR.

We found some results consistent with other French telemedical screening studies: in the Haut-Rhin study reported by Lenoble et al., the rate of all stages of DR was 18.0% (vs. 13.9% in the current study) [13]. However, in this study, patients were recruited differently, referred by their physicians, whereas they were self-referred in our study. Additionally, the duration of diabetes was longer (16.6 vs. 10.0 years), and the glycated hemoglobin level was higher (9.3% vs. 7.1%). In contrast, the 2012 study of Schulze-Döbold et al. in Ile-de-France showed a higher DR rate (24.2%) [20]. However, the population was recruited mostly in hospitals: 17 of the 29 retinal photographers used were from hospitals, and hospital recruitment was described as having a significantly higher number of DRs because of more severe disease in these patients. The study conducted by Soulié-Strougar et al. in 2007 analyzed one of the first DR screening campaigns in Burgundy and found a DR rate of 8.6% [21]. This rate was lower than ours with the same patient recruitment method, but at this time the screening campaign was less well known and selected only those patients most motivated in monitoring their disease, with the best diabetes management, and with a lower DR rate. Finally, in the Italian multicenter NO-BLIND study, in nine public out-patient clinics, a trained diabetologist in each center took photographs of the patients′ fundus oculi using a portable digital ophthalmoscope (MiiS Horus Scope DEC 100) [22]. The prevalence of DR observed was similar to ours (15.5%). These results confirm that telemedicine can rely on adequately trained diabetologists, and it can be performed with portable and less expensive instruments directly at diabetes centers, with results comparable to traditional retinography performed by highly qualified personnel. Therefore, telemedicine is the only cost-saving alternative compared with traditional ophthalmologist examination, regardless of geographical setting, and is well accepted by both patients and providers.

The analysis of the retinal photographs showed pathologies not directly related to diabetes. The prevalence of choroidal nevi is variable, depending on the studies. For example, a 2.1% prevalence of choroidal nevi was found in one Australian study, 5.0% retro-equatorial nevi in an American study, while another Australian study found a 6.5% prevalence [23,24,25]. The current study found the presence of a nevus in 2.8% of the patients. However, we believe that this proportion remains underestimated: photographs were centered on the posterior pole. The prevalence of late AMD in our study was estimated at 3.4%. A 2017 European meta-analysis found the prevalence of late AMD between 0.2% and 5.6% for all age groups [26]. The present study found a proportion within the average range. However, these figures remain challenging to interpret because diabetic patients with known AMD are probably regularly followed up and have no reason to be screened in this way. Here, the prevalence of OHT was 7.8%. A 2006 French cohort study found an 8.9% OHT rate; the two are quite similar [27]. Conversely, it is difficult to estimate the prevalence of glaucoma based only on a photograph of the optic disc, since an abnormality on the optic nerve head is not sufficient to diagnose glaucoma, whereas a normal papilla does not exclude this pathology.

This study is one of the first to compare patients from rural and urban areas in the same cohort. After multivariate analysis, the only difference we found was the age of the patients, who were significantly younger in rural areas. Despite the same duration of diabetes, it is conceivable that patients in rural areas are younger when diagnosed with diabetes, due to poorer access to care and poorer control of diabetes risk factors. This suggests that screening for DR by telemedicine is of interest in rural and periurban areas. It would be interesting to reexamine these figures in a few years when the number of patients in periurban areas will be greater.

As pointed out by Galiero et al. in their recent review of the literature, it is important to be able to maintain screening and follow-up processes in at-risk populations during periods of pandemic and restriction, such as during COVID-19 [28]. The opportunity to safely proceed with the management of diabetic patients by limiting their contact has not only allowed for better management of this disabling complication, but also for inclusion in on-going studies on comorbidities associated with DR in real-life situations [29,30]. Due to the COVID-19 pandemic and the first lockdown, we expect a post-COVID-19 population with poorer diabetes control characteristics or more DR, but this was not the case in the present study, with comparable populations before and after the first lockdown. This lack of a difference may be related to the small sample sizes. This situation has led us to adapt the follow-up of our patients and, most particularly, to develop teleconsultations or therapeutic education protocols [31].

The main strength of this study is the significant number of patients over a long period. Another strength is that it has been around since 2004, resulting in well-organized and standardized recruitment. Moreover, various types of population can participate in this screening: at the beginning, only rural townships were visited, it opened to periurban townships in 2016, and screening in urban townships started in 2021.

The study also has a number of weaknesses, notably a declarative bias from patients: some of them were unable to tell us about their glycated hemoglobin, type of treatment, type of diabetes, or duration of disease, which leads to a lack of information. Over the last year, we have also been impacted by the COVID-19 pandemic, decreasing the number of screened patients. Another of the study’s weaknesses is that this screening campaign only recruits patients voluntarily, which makes it possible that the most assiduous patients were selected.

Areas of improvement can be explored, such as questioning general practitioners (last glycated hemoglobin, treatment), combining visual acuity measurement, performing optical coherence tomography (OCT) of the macula, or using UWF imaging. However, while all of these areas of improvement are important for the early detection and treatment of anomalies, they can be responsible for a significant loss of time and money. The use of automated algorithms is being studied for our screening campaign. We are discussing the possibility of using such an algorithm to identify abnormal photographs that will then be read by an ophthalmologist, although this artificial intelligence technology will allow photographs without noticeable abnormalities to be classified as normal without having been read by an ophthalmologist. Automated detection is now increasingly used and can be combined with the use of UWF imaging to allow easier and faster screening [32].

In conclusion, our screening technique is a meaningful way to detect earlier curable diseases, whether or not they are related to diabetes mellitus, in a population that does not have sufficient ophthalmologic control. However, it has weaknesses and can be improved. This screening campaign has a significant interest in rural and periurban areas. It has been described as cost effective, particularly in rural areas, limiting the transport of patients over long distances, which is reimbursed by healthcare providers in France for patients suffering from diabetes mellitus [33].

## Figures and Tables

**Figure 1 jcm-11-01318-f001:**
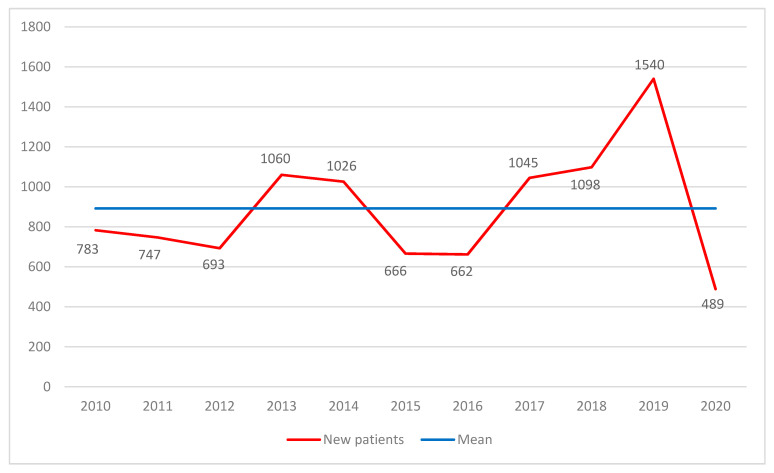
Newly screened patients per year.

**Figure 2 jcm-11-01318-f002:**
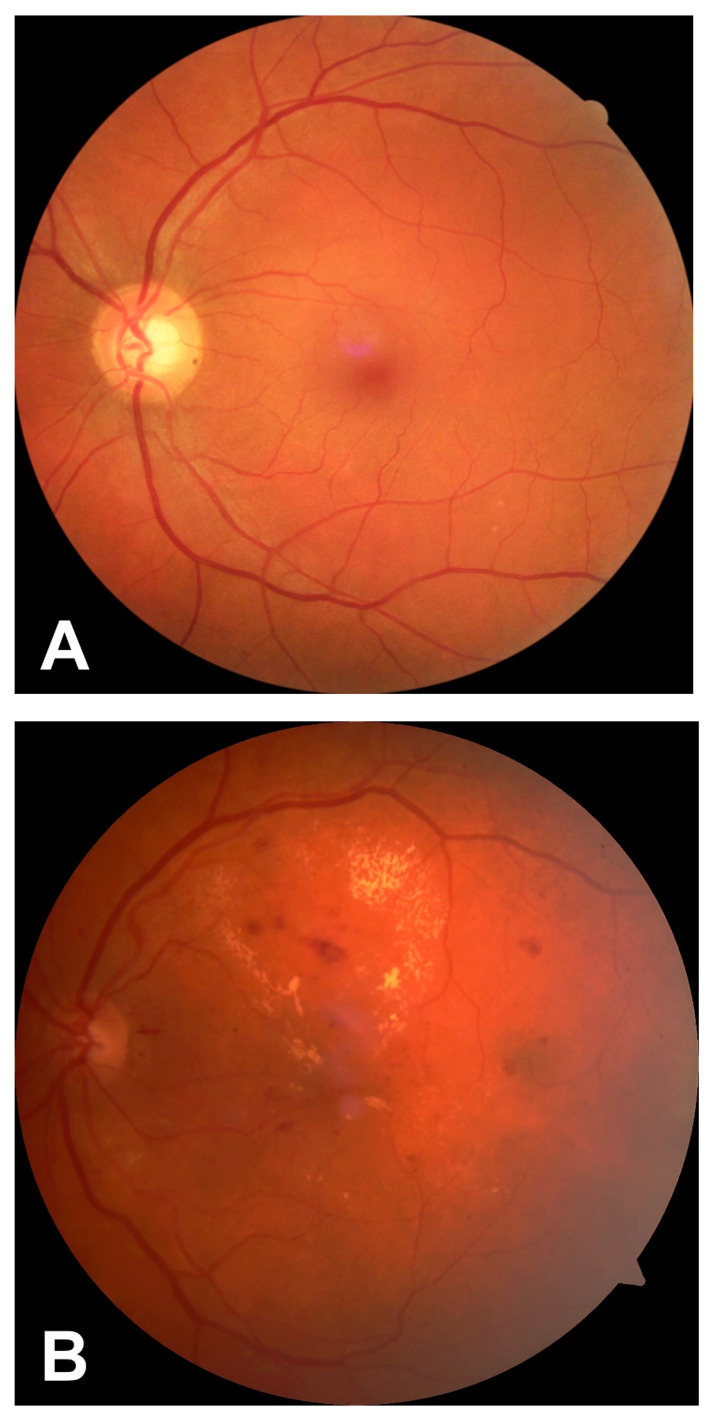
Retinal photographs of patients who underwent roving retinopathy screening. (**A**) Normal left eye fundus; (**B**) Moderate non-proliferative diabetic retinopathy associated with hard exudates of the left eye; (**C**) Lower part of a large pigmented lesion of the right eye.

**Table 1 jcm-11-01318-t001:** Characteristics of patients at their first screening examination.

Gender	
Female	4154 (40.7)
Male	6066 (59.4)
Age (years) ^a^	68.5 (±11.2)
Attending ophthalmologist	3744 (36.5)
Type of diabetes	
Type 1	355 (3.5)
Type 2	8993 (88.0)
Unknown	877 (8.5)
Treatment of diabetes	
Oral antidiabetics	9063 (88.7)
Insulin	1762 (17.2)
Treatment of high blood pressure	6557 (64.2)
Progression ≥ 1 year	1146 (11.2)
Last check-up more than 2 years before	4438 (43.4)
Number of screenings per patient	
1	8321 (81.4)
2	1450 (14.2)
3 or more	449 (4.4)

Data are qualitative, expressed as number (percentage), except for (a) ^a^ quantitative data, expressed as mean (±standard deviation).

**Table 2 jcm-11-01318-t002:** Results of screening at the patients’ first examination.

DR Status	
No DR	8209 (80.3)
All stages	1420 (13.9)
Mild NPDR	948 (9.3)
Moderate NPDR	366 (3.6)
Severe NPDR	84 (0.8)
PDR	22 (0.2)
Unreadable	590 (5.8)
Other pathologies	
Media opacity	692 (6.8)
Late AMD	347 (3.4)
Nevus	287 (2.8)
CRVO	20 (0.2)
OHT	799 (7.8)
Papillary anomaly	418 (4.1)

Data are qualitative, expressed as number (percentage). Abbreviations: DR, diabetic retinopathy; NPDR, non-proliferative DR; PDR, proliferative DR; AMD, age-related macular degeneration; CRVO, central retinal vein occlusion; OHT, ocular hypertension.

**Table 3 jcm-11-01318-t003:** Comparison of patients with and without diabetic retinopathy.

	DR Any Type	No DR	*p*-Value ^a^
Number	1420	8210	
Age (years) ^b^	68.5 (±11.3)	68.3 (±11.2)	0.44
Gender ^c^	841 (59.2)	4859 (59.0)	0.98
Attending ophthalmologist ^c^	474 (33.4)	3041 (37.0)	**<0.01**
HbA1c (%) ^b^	7.44 (±1.4)	6.99 (±1.1)	**<0.001**
Progression of diabetes ^b^	13.76 (±10.9)	9.31 (±8.9)	**<0.001**
Time since last check-up ^b^	3.47 (±3.3)	3.68 (±3.7)	**0.04**
Type of diabetes ^b,d^, 8851 patients			
Type 1	73 (6.1)	258 (3.4)	**<0.001**
Type 2	1129 (93.9)	7391 (96.6)	-
Treatment of diabetes ^b^			
Oral antidiabetics	1181 (83.2)	7380 (89.9)	**<0.001**
Insulin	490 (34.5)	1295 (15.8)	**<0.001**
Other pathologies ^b^			
Media opacity	96 (6.8)	527 (6.4)	0.63
Late AMD	46 (3.2)	294 (3.6)	0.52
Naevus	31 (2.2)	253 (3.1)	0.07
CRVO	2 (0.1)	19 (0.2)	0.76
OHT	131 (9.2)	615 (7.4)	0.02
ONH anomaly	72 (5.1)	337 (4.1)	0.10

^a^ *p*-value patients with DR vs. patients without, bold values denote statistical significance at the *p* < 0.05 level; ^b^ quantitative data: mean (±standard deviation), Welch’s test; ^c^ qualitative data: *n* (%), chi-2 test or Fisher’s test as appropriate; ^d^ exclusion of patients who did not know their diabetes type. Abbreviations: DR, diabetic retinopathy; SD, standard deviation; HbA1c, glycated hemoglobin; AMD, age-related macular degeneration; CRVO, central retinal vein occlusion; OHT, ocular hypertension; ONH, optic nerve head.

**Table 4 jcm-11-01318-t004:** Comparison of the population in rural and suburban townships, univariate analysis.

	Rural Townships	Suburban Townships	*p*-Value ^a^
	2753	1230	-
Age (years) ^b^	73.1 (±11.2)	74.5 (±11.2)	**<0.001**
Males ^c^	1630 (59.2)	738 (60.0)	0.64
Attending ophthalmologist ^c^	910 (33.1)	440 (35.8)	0.09
HbA1c (%) ^b^	7.1 (±1.2)	7.0 (±1.1)	0.06
Treatment of diabetes ^c^			
Oral antidiabetics	2471 (89.8)	1106 (89.9)	0.88
Insulin	467 (17.0)	235 (19.1)	0.10
DR status ^c^			
No DR	2379 (86.4)	1040 (84.6)	0.12
Mild NPDR	220 (8.0)	123 (10.0)	**0.04**
Moderate NPDR	78 (2.8)	42 (3.4)	0.32
Severe NPDR	14 (0.5)	8 (0.7)	0.58
PDR	11 (0.4)	2 (0.2)	0.37
Unreadable ^c^	51 (1.9)	15 (1.2)	0.15
Other pathologies ^c^			
Media opacity	265 (9.6)	106 (8.6)	0.31
Late AMD	118 (4.3)	40 (3.3)	0.12
Nevus	140 (5.1)	48 (3.9)	0.10
CRVO	8 (0.3)	3 (0.2)	1.00
OHT	190 (6.9)	102 (8.3)	0.12
ONH anomaly	117 (4.3)	52 (4.2)	0.97

^a^ *p*-value rural vs. suburban townships, bold values denote statistical significance at the *p* < 0.05 level, ^b^ quantitative data: mean (±standard deviation), Welch’s test; ^c^ qualitative data: *n* (%); chi-square test or Fisher’s test as appropriate. Abbreviations: HbA1c, glycated hemoglobin; DR, diabetic retinopathy; NPDR, non-proliferative DR; PDR, proliferative DR; AMD, age-related macular degeneration; CRVO, central retinal vein occlusion; OHT, ocular hypertension; ONH, optic nerve head.

**Table 5 jcm-11-01318-t005:** Comparison of rural and suburban townships, multivariate analysis.

	Rural Townships	Suburban Townships	Odds Ratio	*p*-Value ^a^
Age (years) ^b^	73.1 (±11.3)	74.5 (±11.2)	0.24 (0.11; 0.51)	**<0.001**
Males ^c^	1630 (59.2)	738 (60.0)	0.95 (0.83; 1.09)	0.46
HbA1c (%) ^b^	7.1 (±1.2)	7.0 (±1.1)	1.10 (1.00; 1.22)	0.06
Insulin ^c^	467 (17.0)	235 (19.1)	0.83 (0.69; 1.00)	0.05
Attending ophthalmologist ^c^	910 (33.1)	440 (35.8)	0.88 (0.77; 1.02)	0.09
No DR ^c^	2379 (86.4)	1040 (84.6)	0.71 (0.40; 1.27)	0.24
Mild NPDR ^c^	220 (8.0)	123 (10.0)	0.79 (0.54; 1.14)	0.20

^a^ *p*-value rural vs. suburban townships, bold values denote statistical significance at the *p* < 0.05 level; ^b^ quantitative data: mean (±standard deviation); ^c^ qualitative data: *n* (%). Abbreviations: HbA1c, glycated hemoglobin; DR, diabetic retinopathy; NPDR, non-proliferative DR.

**Table 6 jcm-11-01318-t006:** Comparison of the study population, before and after the first lockdown.

	Before Lockdown	After Lockdown	*p*-Value ^a^
	367	441	
Age (years) ^b^	70.3 (±11.7)	69.1 (±10.2)	**0.04**
Males ^c^	212 (57.8)	271 (61.5)	0.31
HbA1c (%) ^b^	7.1 (±1.2)	6.9 (±1.0)	0.28
Progression of diabetes (years) ^b^	11.9 (±10.4)	11.2 (±9.6)	0.37
Insulin-dependent ^c^	71 (20.5)	69 (16.8)	0.22
DR status ^c^			0.56
No DR	325 (88.5)	393 (89.1)	-
Mild NPDR	29 (7.9)	37 (8.4)	-
Moderate NPDR	7 (1.9)	9 (2.0)	-
Severe NPDR	5 (1.4)	2 (0.5)	-
PDR	1 (0.3)	0	-

^a^ *p*-value before vs. after the lockdown, bold values denote statistical significance at the *p* < 0.05 level; ^b^ quantitative data: mean (±standard deviation); ^c^ qualitative data: *n* (%). Abbreviations: HbA1c, glycated hemoglobin; DR, diabetic retinopathy; NPDR, non-proliferative DR; PDR, proliferative DR.

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
