# Peer review of "Mobile Telemedicine Screening for Diabetic Retinopathy Using Nonmydriatic Fundus Photographs in Burgundy: 11 Years of Results"

_jcm, 2022, doi:10.3390/jcm11051318_

Round 1
Reviewer 1 Report
The authors analyzed the results of mobile screening for DR by retinophotographs, and compared the results between rural and peri-rural areas. They insisted that their screening technique is a meaningful way to detect earlier curable diseases, linked to diabetes or not.
The writing is somewhat distracting that it is difficult to grasp the messages the authors tried to deliver. They were trying to contain too much content at once, and some paragraphs contain content that has no relationship with the title. It would be better to focus on the topic authors want to convey more clearly. Additionally, the authors should show a representative photo image they used for analyses.
Author Response
We thank the reviewers for their careful review of our manuscript entitled “Mobile telemedicine screening for diabetic retinopathy using nonmydriatic fundus photographs in Burgundy: 11 years of results” [jcm-1591851].
Please note that all the material provided has been edited by a professional English-speaking copy-editor (Linda Northrup, linda.northrup@orange.fr).
Below you will find our responses and modifications.
Reviewer 1
- The writing is somewhat distracting that it is difficult to grasp the messages the authors tried to deliver. They were trying to contain too much content at once, and some paragraphs contain content that has no relationship with the title. It would be better to focus on the topic authors want to convey more clearly.
We thank reviewer 1 for this comment. We could incorporate section 3.3 on comparing patients with diabetic retinopathy to those without in section 3.2 of the screening results. However, we feel it is important to include this information because 1) the purpose of this mobile screening is to detect diabetic complications in asymptomatic patients; 2) this is a point of interest for the reader and for health policies that will be able to adapt their patient profiling. If the reviewer finds any paragraphs that do not seem relevant to the topic, or that are unclear, now that the English and the fluency of the text have been edited, we would be happy to make the appropriate changes.
- Additionally, the authors should show a representative photo image they used for analyses.
You will find in Figure 2 retinal photographs of patients who underwent roving retinopathy screening. in A - Normal left eye fundus, B - Moderate non-proliferative diabetic retinopathy associated with hard exudates of the left eye, C - Lower part of a large pigmented lesion of the right eye. Line 180, 186.
These figures will allow the reader to visualize the contribution of the techniques described.
Respectfully,
Florian BAUDIN MD
Reviewer 2 Report
The paper is interesting. However, this reviewer raises few issues that need to be addressed.
1- In this retrospective, cross-sectional study based on an itinerant telemedicine screening performed between September 2009 and November 2020 in Burgundy, France, the authors observed a prevalence of 13.9% of diabetic retinopathy. Fundus retinophotographs were realized using a Canon NM CR-2 Plus Fundus Camera (Canon, Tokyo, Japan), a classic fundus camera, by trained orthoptist.
In the Italian multicenter study NO-BLIND, in nine public out-patients clinics for 6 months in 2017, a carefully trained diabetologist in each center took photographs of the patients' fundus oculi through a portable digital ophthalmoscope (MiiS Horus Scope DEC 100, Digital Eye ‐ fundus Camera , Medimaging Integrated Solution Inc Brussels, Belgium). The observed prevalence of DR was 15.5%, as assessed by a trained eye doctor at a single reading center who received all photographs (Diabetes Metab Res Rev. 2019; e3113. Doi: 10.1002 / dmrr.3113.) These results confirm that telemedicine can involve also adequately trained diabetologists, and it can be performed with portable and less expensive instruments directly at the territorial diabetes centers with results comparable to traditional retinography, performed by highly qualified personnel. Therefore, telemedicine represents the only cost-saving alternative compared with traditional ophthalmologist examination, regardless of geographical setting, and is well accepted by both patients and providers. This problem and previous reference should be added in the manuscript.
2- The use of telemedicine has always found a natural candidate in the elderly, even more so during the COVID-19 pandemic (J Diabetes Res. 2020 Oct 14;2020:9036847. doi: 10.1155/2020/9036847). The possibility of increasing in this way the number of diabetic patients, especially the elderly, who were able to carry out both the screening and the follow-up of DR, not only allowed a better management of this disabling complication of diabetes, but also to expand the studies on comorbidities associated with RD in real life setting (Nutr Metab Cardiovasc Dis. 2019 Sep;29(9):923-930. doi: 10.1016/j.numecd.2019.05.065. - Diabetes Res Clin Pract. 2019 Apr;150:236-244. doi: 10.1016/j.diabres.2019.03.028.). These issues and above references should be discussed in the paper.
3- The paper should be reviewed by a native English speaker.
Author Response
We thank the reviewers for their careful review of our manuscript entitled “Mobile telemedicine screening for diabetic retinopathy using nonmydriatic fundus photographs in Burgundy: 11 years of results” [jcm-1591851].
Please note that all the material provided has been edited by a professional English-speaking copy-editor (Linda Northrup, linda.northrup@orange.fr).
Below you will find our responses and modifications.
Reviewer 2
- In this retrospective, cross-sectional study based on an itinerant telemedicine screening performed between September 2009 and November 2020 in Burgundy, France, the authors observed a prevalence of 13.9% of diabetic retinopathy. Fundus retinophotographs were realized using a Canon NM CR-2 Plus Fundus Camera (Canon, Tokyo, Japan), a classic fundus camera, by trained orthoptist. In the Italian multicenter study NO-BLIND, in nine public out-patients clinics for 6 months in 2017, a carefully trained diabetologist in each center took photographs of the patients' fundus oculi through a portable digital ophthalmoscope (MiiS Horus Scope DEC 100, Digital Eye ‐ fundus Camera , Medimaging Integrated Solution Inc Brussels, Belgium). The observed prevalence of DR was 15.5%, as assessed by a trained eye doctor at a single reading center who received all photographs (Diabetes Metab Res Rev. 2019; e3113. Doi: 10.1002 / dmrr.3113.) These results confirm that telemedicine can involve also adequately trained diabetologists, and it can be performed with portable and less expensive instruments directly at the territorial diabetes centers with results comparable to traditional retinography, performed by highly qualified personnel. Therefore, telemedicine represents the only cost-saving alternative compared with traditional ophthalmologist examination, regardless of geographical setting, and is well accepted by both patients and providers. This problem and previous reference should be added in the manuscript.
We thank Reviewer 2 for this extremely relevant addition. You will find on lines 323–332 the addition below incorporating these remarks and the above-mentioned reference.
“Finally, in the Italian multicenter NO-BLIND study, in nine public out-patient clinics, a trained diabetologist in each center took photographs of the patients' fundus oculi using a portable digital ophthalmoscope (MiiS Horus Scope DEC 100) [1]. The prevalence of DR observed was similar to ours (15.5%). These results confirm that telemedicine can rely on adequately trained diabetologists, and it can be performed with portable and less expensive instruments directly at diabetes centers, with results comparable to traditional retinography, performed by highly qualified personnel. Therefore, telemedicine is the only cost-saving alternative compared with traditional ophthalmologist examination, regardless of geographical setting, and is well accepted by both patients and providers.”
- The use of telemedicine has always found a natural candidate in the elderly, even more so during the COVID-19 pandemic (J Diabetes Res. 2020 Oct 14;2020:9036847. doi: 10.1155/2020/9036847). The possibility of increasing in this way the number of diabetic patients, especially the elderly, who were able to carry out both the screening and the follow-up of DR, not only allowed a better management of this disabling complication of diabetes, but also to expand the studies on comorbidities associated with RD in real life setting (Nutr Metab Cardiovasc Dis. 2019 Sep;29(9):923-930. doi: 10.1016/j.numecd.2019.05.065. - Diabetes Res Clin Pract. 2019 Apr;150:236-244. doi: 10.1016/j.diabres.2019.03.028.). These issues and above references should be discussed in the paper.
We thank the reviewer for the suggestions to add the work of the Sasso et al.’s team, who have extensive experience in screening for diabetic retinopathy and diabetes-associated conditions. We therefore propose to add lines 360-366:
“As pointed out by Galiero et al. in their recent review of the literature, it is important to be able to maintain screening and follow-up processes in at-risk populations during periods of pandemic and restriction such as during COVID-19 [2]. The opportunity to safely proceed with the management of diabetic patients by limiting their contact has not only allowed for better management of this disabling complication, but also for inclusion in on-going studies on comorbidities associated with DR in real-life situations [3,4].”
- The paper should be reviewed by a native English speaker.
We have followed your advice and had the article edited by a professional English-speaking copy-editor (Linda Northrup, linda.northrup@orange.fr).
References
- Sasso, F. C.; Pafundi, P. C.; Gelso, A.; Bono, V.; Costagliola, C.; Marfella, R.; Sardu, C.; Rinaldi, L.; Galiero, R.; Acierno, C.; de Sio, C.; Adinolfi, L. E., Telemedicine for screening diabetic retinopathy: The NO BLIND Italian multicenter study. Diabetes Metab. Res. Rev. 2019, 35 (3), e3113.
- Galiero, R.; Pafundi, P. C.; Nevola, R.; Rinaldi, L.; Acierno, C.; Caturano, A.; Salvatore, T.; Adinolfi, L. E.; Costagliola, C.; Sasso, F. C., The Importance of Telemedicine during COVID-19 Pandemic: A Focus on Diabetic Retinopathy. J Diabetes Res 2020, 2020, 9036847.
- Sasso, F. C.; Pafundi, P. C.; Gelso, A.; Bono, V.; Costagliola, C.; Marfella, R.; Sardu, C.; Rinaldi, L.; Galiero, R.; Acierno, C.; de Sio, C.; Caturano, A.; Salvatore, T.; Adinolfi, L. E., High HDL cholesterol: A risk factor for diabetic retinopathy? Findings from NO BLIND study. Diabetes Res. Clin. Pract. 2019, 150, 236-244.
- Sasso, F. C.; Pafundi, P. C.; Gelso, A.; Bono, V.; Costagliola, C.; Marfella, R.; Sardu, C.; Rinaldi, L.; Galiero, R.; Acierno, C.; Caturano, A.; de Sio, C.; De Nicola, L.; Salvatore, T.; Nevola, R.; Adinolfi, L. E.; Minutolo, R., Relationship between albuminuric CKD and diabetic retinopathy in a real-world setting of type 2 diabetes: Findings from No blind study. Nutr. Metab. Cardiovasc. Dis. 2019, 29 (9), 923-930.
Respectfully,
Florian BAUDIN MD
Round 2
Reviewer 1 Report
The authors reinforced the points appropriately.
Author Response
We thank the reviewer for his time and consideration